mathematical modelling

COVID-19, phone calls, correlation, early-alarm, live-tracking

**Author for correspondence:**
Ezequiel Alvarez
e-mail: sequi@unsam.edu.ar

# Estimating COVID-19 cases and outbreaks on-stream through phone calls

Ezequiel Alvarez[1], Daniela Obando[2], Sebastian Crespo[2], Enio Garcia[2], Nicolas Kreplak[2] and Franco Marsico[2]

[1]International Center for Advanced Studies (ICAS), ICIFI-CONICET ECyT-UNSAM, Campus Miguelete, 25 de Mayo y Francia, CP1650, San Martìn, Buenos Aires, Argentina
[2]Ministerio de Salud de la Provincia de Buenos Aires, La Plata, Buenos Aires, Argentina

 EA, 0000-0001-6780-4415

One of the main problems in controlling COVID-19 epidemic spread is the delay in confirming cases. Having information on changes in the epidemic evolution or outbreaks rise before laboratory-confirmation is crucial in decision making for Public Health policies. We present an algorithm to estimate on-stream the number of COVID-19 cases using the data from telephone calls to a COVID-line. By modelling the calls as background (proportional to population) plus signal (proportional to infected), we fit the calls in Province of Buenos Aires (Argentina) with coefficient of determination $R^2 > 0.85$. This result allows us to estimate the number of cases given the number of calls from a specific district, days before the laboratory results are available. We validate the algorithm with real data. We show how to use the algorithm to track on-stream the epidemic, and present the Early Outbreak Alarm to detect outbreaks in advance of laboratory results. One key point in the developed algorithm is a detailed track of the uncertainties in the estimations, since the alarm uses the significance of the observables as a main indicator to detect an anomaly. We present the details of the explicit example in Villa Azul (Quilmes) where this tool resulted crucial to control an outbreak on time. The presented tools have been designed in urgency with the available data at the time of the development, and therefore have their limitations which we describe and discuss. We consider possible improvements on the tools, many of which are currently under development.

## 1. Introduction

The COVID-19 epidemic has been causing global damage to practically all aspects of world society since early 2020. Although a huge effort in many fields of sciences is being made

to mitigate its effects, the disease is continuously spreading and, in many regions, a second wave is causing great concern. The difficulties in controlling the epidemic are in part due to a crucial combination of being highly contagious [1], having a long incubation period [2] during which infections are possible a few days before symptoms onset [3], having mild or asymptomatic cases [1] and also because the diagnosis may take a few days after contacting the Health Care system. In particular, the latter yields that outbreaks spread and epidemic evolves while laboratory results are being processed. This effect being more important in low- and medium-income countries due to operational and logistic problems, generally caused by technological and economic inequalities [4,5].

We present in this work a method to mitigate the epidemic effects by estimating the number of COVID-19 cases without having to wait for laboratory confirmations. This provides the Health Care system with a tool to react in advance and evaluate current or next Public Health policies.

In mass accidents or major catastrophes, early warning systems (EWS) play a key role for disaster mitigation [6–8] decreasing response times and improving their effectiveness. The main strategy of EWS in infectious disease surveillance is the incorporation of information produced nearly from the infection [9–11]. In this case, the symptoms onset and their detection by the individual and community health systems is the first detectable signal of cases and, in particular, an outbreak. EWS based on syndromic surveillance have been applied in epidemiological surveillance for early outbreaks identification and confirmation [12–16]. One of the main characteristics of EWS is the utilization of health information provided by the population in order to activate local alarms. Nowadays, with the wide use of cell phone applications and specific health-system phone lines, important databases with information about syndromic surveillance are generated each day [17]. Geo-location plays a main role in spatial and temporal definition of the outbreaks detected by EWS [18].

In Buenos Aires Province (Argentina), the COVID-19 phone line 148 is one of the first contacts between a person that believes themselves to be infected and the Health Care system. The trained Health Care team receives and responds to people's questions generating, simultaneously, a syndromic surveillance database. If the person has symptoms that could indicate a COVID19 infection, they are instructed to follow the corresponding protocol. Importantly, such a syndromic database was used as an input for estimation of cases and outbreak detection in Buenos Aires Province.

This work is divided as follows. In §2, we describe the COVID-line data and present the details of the mathematical model to estimate the number of cases using the phone calls data. In §3, we show how the model works in Buenos Aires Province and how it can be used to track on-stream the epidemic. In §4, we present the Early Outbreak Alarm and show its details in Villa Azul (Quilmes) case. We discuss the limitations and current improvements of the model in §5, and we present our conclusions in §6.

# 2. Estimating on-stream COVID-19 cases through calls to a COVID-line

We describe the mathematical model implemented to relate phone calls to a COVID-line to laboratory-confirmed cases per district per day. In the following paragraphs, we outline the functioning of the 148 COVID-line and then we describe the details of the model.

## 2.1. COVID-line 148 in Buenos Aires Province

Buenos Aires Province (PBA for its acronym in Spanish) is the most populated province of Argentina, with more than 17 million inhabitants. Around 13 million people live in the Metropolitan area surrounding the City of Buenos Aires. Importantly, the remaining 4 million live in the vast area with low population density known as the interior of the province. This demographic heterogeneity leads to a hyper-centralized Health Care system. In order to attend to the growing demand for medical assistance caused by COVID-19, public health authorities implemented in February 2020 a COVID-specific phone line that is reached by dialing 148. The objective of this COVID-line is to address all community concerns related to COVID, which include questions, doubts, symptom reports and reference to the Health Care system, among others.

The staffing of the COVID-19 phone line was increased as the epidemic spread in PBA. The call request grew from a few hundreds per day in March up to approximately 20k per day at the end of August. Until late June the system was able to meet the demand, and all calls requiring assistance were taken. We therefore consider that during this regime an indicator coming from this COVID-line would be relatively unbiased. This is specially true if one compares this to other indicators as testing, or laboratory processing, which were changing their behaviour considerably as epidemic spread during this period. We find that 1 April to 26 June is a period in which the COVID-line has been relatively stable.

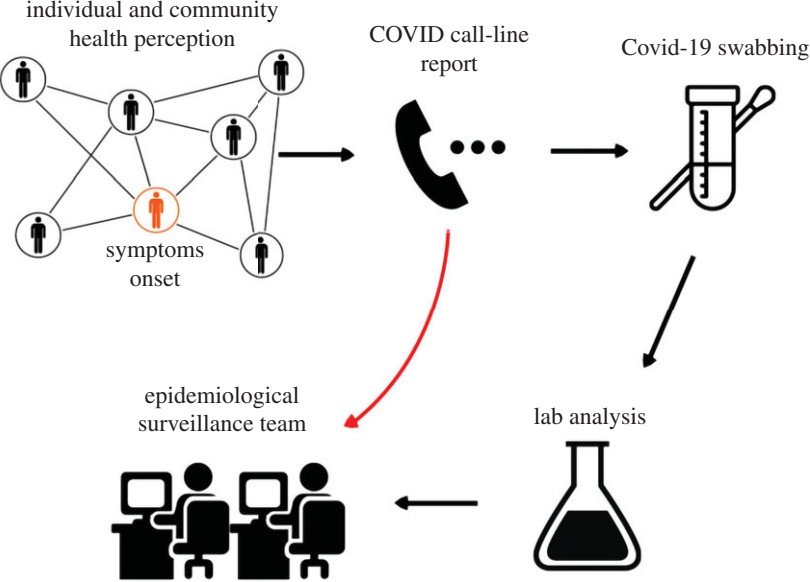

**Figure 1.** COVID-line 148 workflow. As people call the COVID-line upon their health perception, the COVID-trained operators determine whether they correspond to suspicious or close-contact case. In such a case, their record is passed to the epidemiological surveillance team and a COVID-19 swabbing is ordered. Some days later the swabbing laboratory result is added to the corresponding record. The algorithm described in this paper works with the first part of the information which is delivered on-stream as the operators determine the case passes the corresponding threshold.

As people call the COVID-line 148, they enter into an automatic voice menu in which one of the options corresponds to COVID-like symptoms. As users go into this option their call is taken by a COVID-trained operator and a short questionnaire on their experience indicates whether the call does not pass the threshold to be registered or corresponds to one of the two registered categories: close-contact and suspicious case. If the call corresponds to any of these categories, then the operator registers their data and in particular the district from which they are calling. We depict in figure 1 the workflow of the COVID-line. At the early stage that the system was implemented, the record did not contain reliable information on the exact address of the user. This crucial fact led us to develop the system we explain below by restricting our information on the user to only their district. Although future upgrades of the system are providing more accurate location of the call, the current work restricts to the caller district and only once their call is taken by a COVID-trained operator.

## 2.2. Mathematical model to estimate cases from phone calls to the 148 COVID-line

We present the mathematical model to estimate the new infected using the phone call data, and apply it to PBA. The reasoning in this section follows the same lines as in [19], but with different purposes and different filtering in the dataset.

We consider a dataset of calls from many districts and during a given time range to a COVID-line. Each one of these calls can either be

> background: people with similar symptoms but not infected; or
> signal: people infected with COVID-19.

Under reasonable assumptions of homogeneity in space and time we can model that background calls in each district and time-window are proportional to the total district population and the time-window length. Whereas signal calls are proportional to the total number of infected people in the district whose record is opened in the corresponding time-window, even though their laboratory-confirmation may be available in a later time. Therefore, if we divide all our dataset in chunks corresponding in space to the districts in PBA, and in time to time-windows of $\Delta t^{(j)}$ days that can be arbitrarily chosen, we can pose the following equations for all the chunks labelled by $j$:

$$n_c^{(j)} = \theta_p \, \Delta t^{(j)} \, N_p^{(j)} + \theta_I \, N_I^{(j)}, \tag{2.1}$$

where $N_p^{(j)}$ is the population of the corresponding district and $N_I^{(j)}$ is the number of confirmed infected at the same district and whose record was opened during the corresponding time-window. On the left-hand side, $n_c^{(j)}$ is the fit to the total number of calls, whereas $N_c^{(j)}$ (not in the equation) is the total number of actually placed calls. Observe, therefore, that this set of equations (one for each chunk $j$) can be extended depending on the chosen time-window length. Once this set of $j = 1 \ldots k$ equations has been posed, we can fit the best values of coefficients $\theta_{p,I}$ that minimize the square distance between $n_c^{(j)}$ and $N_c^{(j)}$. We stress that there are only two coefficients $(\theta_{p,I})$ that must fit all $k$ different equations for each chunk.

This fit works better if all chunks correspond to periods in which the testing methods have not changed drastically, as it can be for instance if the number of daily tests is modified considerably, or if new symptoms are considered as thresholds for testing, among others. The reason for requiring this is to have a coherent balance between the number of infected reported and the number of calls in all chunks all the time. With this objective, is better to re-fit the parameters every time there are major changes in the testing and reporting methods. It is also important to observe at this point that in using the same $\theta_{p,I}$ to fit regions with very different sociocultural aspects may lead to eventual biases. Therefore, if the dataset is large enough, it may be convenient in some cases to fit different $\theta_{p,I}$ for each region and use many time windows for the fit. In such a case, these variations of $\theta_{p,I}$ along the regions may provide insightful sociocultural information. In choosing how to do the fit, one is balancing between reducing relative statistical uncertainty with many chunks, or reducing sociocultural biases by separating the regions. Along this work, since we have a limited dataset, we have fitted all regions with the same $\theta_{p,I}$ parameters, accepting that a slight bias may be introduced in the outcome.

Once the parameters $\theta_{p,I}$ in equation (2.1) have been fitted, including their uncertainty from the fit, we can estimate the number of new infected in a given chunk as

$$ n_I^{(j)} = \frac{1}{\theta_I} \left( N_c^{(j)} - \theta_p \, \Delta t^{(j)} \, N_p^{(j)} \right). \tag{2.2} $$

Observe that the right-hand side requires data that is obtained in the same day, and therefore one can estimate the number of cases $n_I$ on-stream, without need of waiting for the laboratory results. Note, that the algorithm allows the estimation of the total number of new cases in each chunk, but not the determination of *which* of the calls correspond to the new cases. The uncertainty in the estimation of new infected in each chunk, $\xi_{n_I^{(j)}}$, is computed by applying the usual error expansion formula on equation (2.2). If variables are correlated, as for instance $\theta_p$ and $\theta_I$, one should take this into account; however, in our case we neglected this correlation in comparison to other terms. For the parameters $\theta_{p,I}$, we use the uncertainty coming from the fit, for $N_c$ we use Poisson uncertainty, and for $N_p$ one should decide whether to add a systematic uncertainty or only use Poisson, as we did in this work. Observe that since Poisson uncertainty for $N_{c,p}$ goes as $\sqrt{N_{c,p}}$, then the absolute uncertainty $\xi_{n_I^{(j)}}$ will increase as $n_I$ increases. However, the relative uncertainty $\xi_{n_I^{(j)}}/n_I$ decreases when the number of estimated cases increase, which makes the predictions more robust to fluctuations in terms of their relative size. This and other related effects are explicitly seen in the figures of the examples discussed in the following sections. As discussed below, uncertainties in the estimations play a central role in the design of the Early Outbreak Alarm, and therefore should be handled with care, especially the systematic ones, if present.

In order to apply this algorithm in PBA we have used the dataset of phone calls to the 148 COVID-line. We work with all the phone calls entering the COVID-line that reach the threshold for being close contact or suspicious case. The reason for this filtering is because the district the user is calling from is registered by the operator. Although the address is also in principle registered on most occasions, in practice, many ambiguities, misspelled words, or other unintended errors yield that only approximately 50%–70% of the times it can be correctly reconstructed. We consider the dataset of calls between 1 April and 26 June only, because after this date the call centre was overloaded and all the calls could not be taken, yielding intractable biases. For this period, we have fitted the data a few times in different datasets, obtaining fairly similar results and with coefficient of determination always satisfying $R^2 > 0.85$. In particular, as cases were increasing, we were obtaining more accurate estimations for $\theta_I$, as can be expected.

In order to show the robustness of the hypotheses, we show how this model works with data from 1 May to 26 June, divided in two equal length time-windows each. We consider all districts in PBA whose number of calls in these chunks is greater than 100. After this filtering we are left with 43 chunks, i.e. 43 data points. After performing the fit indicated in equation (2.1) we obtain

$$ \theta_p = (5.16 \pm 1.59) \times 10^{-6} \text{ calls per inhabitant per day} $$

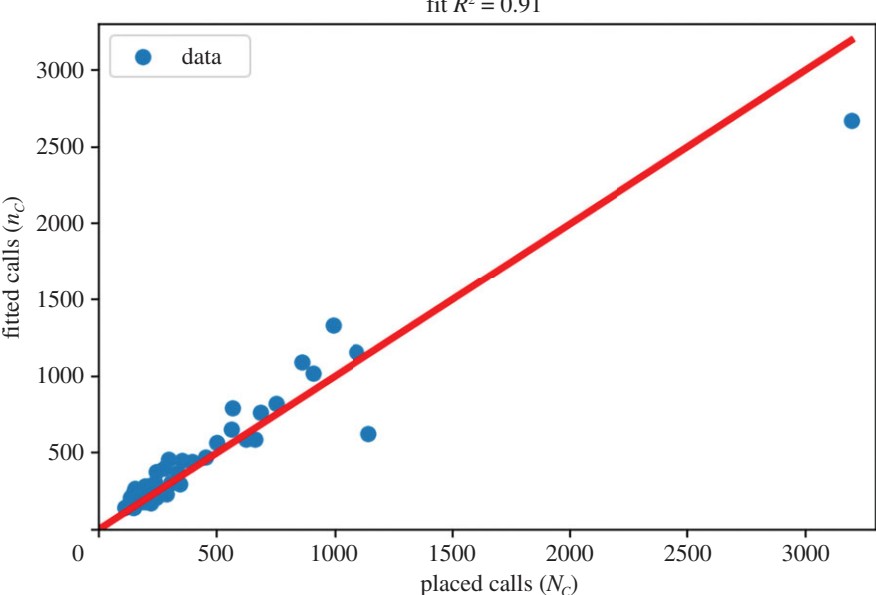

**Figure 2.** Placed versus fitted calls during the fitted period (1 May to 26 June divided in two time-windows). The number of fitted calls comes out from the number of laboratory-confirmed COVID cases using the fit in equation (2.1). The upper right data-point corresponds to La Matanza district whose population of 1.7 million inhabitants is approximately at least three times larger than all other districts.

and

$$\theta_I = 0.69 \pm 0.05 \text{ calls per infected people.}$$

It is worth noticing that the precise values of these fitted coefficients have a strong dependence on the process of call filtering and call system architecture. In particular, these values differ from those in [19] because we are considering a different level of filtering to obtain the district of each user. The fit for this dataset yields a coefficient of determination $R^2 = 0.91$, which indicates the robustness of the involved hypotheses. In figure 2, we show the comparison between data and fit for the number of phone calls, as posed in equation (2.1).

# 3. Tracking epidemic through model estimations

The mathematical model described in the previous section provides a framework to estimate many days in advance the number of laboratory-confirmed cases per day, as a function of the spatio-temporal distribution of phone calls to the COVID-line. This is a compelling achievement because the phone call information is available on-stream, whereas laboratory confirmation of cases may require from a few days to up to a week after patients report their first symptoms. Along this section we show how this system can be used to have an estimate of the epidemic evolution on-stream, along with real case results in PBA.

As this system was developed there was no time for validation. However, obtaining a very satisfactory $R^2 \gtrsim 0.85$ in the fit was a signal that the model was insofar working well. As months went by, we had the possibility of comparing in a long range time-window the model estimation against the measured laboratory-confirmed number of cases per day per district. In figure 3, we show the comparison between the estimation and the late laboratory-confirmed cases per day for any two districts in PBA. Similar results are obtained for other districts. It is central to observe in this figure that the number of laboratory-confirmed cases (red line) is information that is available many days after the corresponding date, whereas the model estimation (blue) is available at the end of each day. As can be seen in the figure, the estimation has a good agreement with the real data. There are a few date ranges in which door-to-door swabbing through *DETECTAR* operatives [20] induce an expected sub-estimation in cases.

This syndromic surveillance has been used to follow the size, spread and tempo of outbreaks, to monitor disease trends and to provide reassurance that a potential outbreak has not occurred.

**Figure 3.** Comparison of real data (red) versus model estimation (blue) for two example districts of Buenos Aires Province (La Matanza and Lomas de Zamora). Recall that the red line corresponding to the real confirmed cases, with their record opening in the corresponding date, is reconstructed many days later. In the dates in which the red line goes above the estimation, it is usually because *DETECTAR* operatives (door-to-door testing [20]) were carried out. In general, the model yields a very good estimation to monitor the epidemic in all affected PBA districts.

In particular, it has also been very useful as an early outbreak detection, as we detail in the next section. Syndromic surveillance systems seek to use existing health data in real time to provide immediate analysis and feedback to those in charge with investigation and follow-up of potential outbreaks. Particularly, the data collected by the COVID-line calls proved to be a valuable and reliable input to track the epidemic along the PBA.

The tracking of the epidemic through this model is especially useful when the capacity overload of the diagnostic centres leads to delays in obtaining results. For this reason, having a real-time and relatively unbiased estimation of cases gives the Public Health authorities the possibility of taking actions in time [14]. Furthermore, in a disaster scenario prioritization, this tool takes a main role when resources and time are limited, as occurred at the end of June in PBA. The calls-based syndromic surveillance allowed a rapid characterization of the different PBA districts in terms of their epidemiological status, and the consequent action was taken in order to mitigate the epidemic effects.

# 4. Early Outbreak Alarm

In this section, we detail a compelling by-product of the model in §2 to detect COVID-19 outbreaks considerably earlier than through laboratory confirmation. We briefly describe the working of the model and then provide its detail through the description of a real case that occurred in mid-May in Villa Azul (Quilmes) and Villa Itatí (Avellaneda) in PBA.

## 4.1. Identifying an outbreak formation

Using the on-stream estimation of the cases per day in each district, we are interested in developing a statistical and automatic tool that can trigger an alarm when a potential outbreak is on the rise. Having an early alarm on this kind of epidemiological feature is a crucial tool to avoid its spread and drastic consequences.

To detect a potential outbreak there are many indicators that should be simultaneously analysed. On one hand it is important to have an estimation of the daily absolute and relative number of cases and, on the other hand, it is also important to have an estimation on the daily variation of these observables. To have an objective quantitative indicator of the potential of an outbreak in a given region, it is essential to have a correct assessment of the uncertainties in all the estimations of the model. As the implemented system is intended to be an Early Outbreak Alarm, we have considered that the important indicator is the *significance* of all the basic indicators which signal an anomaly as they depart from zero. Here, significance is defined as the distance to zero from the central value of the indicator, measured in units of its uncertainty. Or, in other words,

$$\text{significance} = \frac{\text{central value}}{\text{uncertainty}}. \tag{4.1}$$

As can be seen in equation (4.1), the correct computation of the uncertainties (or error bars) is crucial for the functioning of the Early Outbreak Alarm.

The developed algorithm computes every day the estimation for the total number of new cases in each district in PBA. Since in the studied time-window, specially before June, the number of estimated cases per day of many districts was approximately 5–10, we considered to include the estimation of cases for the last 2 days. This would reduce the relative Poisson uncertainty due to small numbers. We computed the number of estimated cases in absolute value, and also relative to 100 000 inhabitants to be equally sensitive to all districts.

A third and decisive observable that signals the level of danger of an outbreak is the daily increase of estimated cases. Given the daily estimation provided by the mathematical model, we can recognize a rapidly increasing curve in many ways. We have chosen to fit a straight line to the case estimation for the last 3 days and use the slope of this line as an estimation of the central value of the daily increase. We also use the significance as the most relevant indicator to decide the level of danger of each district. In this case, the computation of the error bar in the slope of the line includes all the uncertainties in each day estimation included in the computation of the fit uncertainty through the least-squares residuals. We use 3 days to fit a line because it is the minimum time needed to see a 2-day consecutive increase, while being still well ahead of the laboratory results. In addition, 3 days is also a good time-window for the specific COVID-19 characteristics.

This Early Outbreak Alarm has provided the PBA Health Care administration with very important tools to identify possible outbreaks during the rise of the epidemic curve. Since the granularity of the algorithm is very poor (districts), the system needs to be complemented with other independent indicators, in particular those which can help to provide a more accurate location of the outbreak. This was usually done by calling back manually the recorded cases, and then by sending *DETECTAR* operatives [20] to verify if in fact the *in situ* conditions would be as predicted. The Early Outbreak Alarm has indicated many outbreaks that have been controlled between mid-April and mid-June. In particular, we describe in the following paragraphs the very special[1] case of Villa Azul (Quilmes) and provide the details on how the Early Outbreak Alarm indicated the Quilmes district.

## 4.2. Case study: Villa Azul, Quilmes

We report the details of one of the outbreaks indicated by the Early Outbreak Alarm in mid-May in Quilmes district. This case was the first major outbreak in a low-income neighbourhood in PBA and had a great impact in the news [21], not only for its magnitude but also because of its early detection that drove a strict lock-down and isolation of the outbreak to control its spread to the close neighbourhoods.

On 20 May, the alarm was indicating a large number of estimated cases in Quilmes district (figure 4); in particular, Quilmes had the top estimation in number of cases per inhabitants of the last 2 days, as measured through the significance of the indicator. In addition to this, the indicator of the daily increase fit was also indicating Quilmes as the top district in significance (figure 5). This last indicator on the daily increase fit to the last 3 days can be visualized in figure 6*a*, where we plot the daily estimation for the last 7 days on Quilmes. The fit is obtained using the last three data points in red. Given all these indicators pointing to Quilmes district, the surveillance team took the duty to locate the phone calls and observed an excess coming from Villa Azul, a low-income neighbourhood in Quilmes and next to Avellaneda district.

These observations indicated in advance by the Early Outbreak Alarm needed to be verified by an independent complementary indicator. On the following day, a *DETECTAR* operative [20] was sent to Villa Azul, where the acute situation was verified and door-to-door swabbing with urgent laboratory results started right away. As first results were confirming the outbreak in Villa Azul, the PBA Administration decided on a strict lock-down and isolation for 14 days from 24 May [21].

## 4.3. Villa Azul epidemiological and operational description

Villa Azul (Quilmes) and Villa Itatí (Avellaneda) are two adjacent low-income neighbourhoods. The last demographic analysis indicates that Villa Azul has a population of 3128 and Villa Itatí 15 142. High-density building and housing, and tiny streets bring the population into close contact. These characteristics make these neighbourhoods susceptible to a fast spread [22]. Taking this into account,

---

[1]This case covered the headlines in the news for several weeks [21].

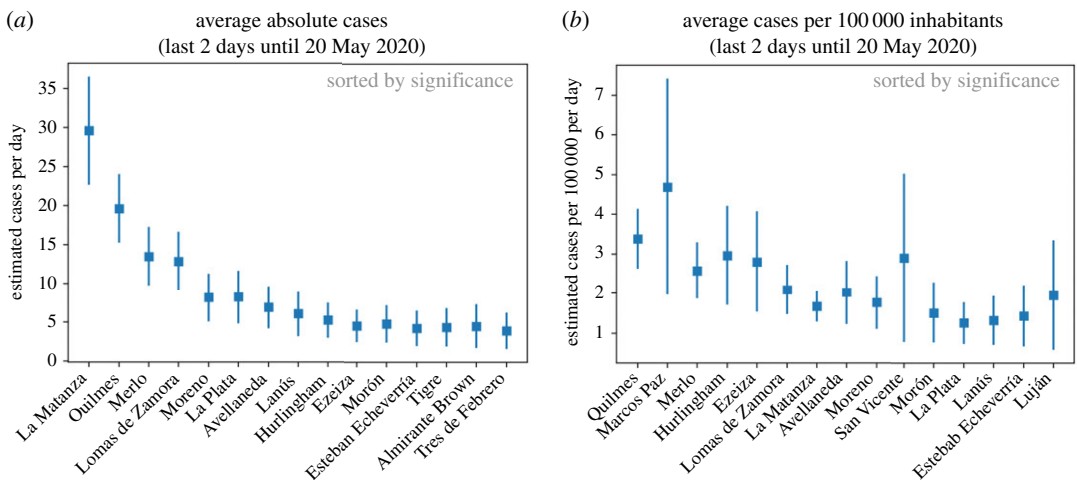

**Figure 4.** Cases-per-day estimation using the model on the COVID-line phone calls for the last 2 days. Observe that districts are not sorted by their central value, but by their significance, which is defined as the rate between the central value and the uncertainty. This is why error bars are crucial to provide an Early Outbreak Alarm. Results are shown in absolute value (*a*) and relative to every 100 000 inhabitants (*b*). In the figure, we show the scenario for 20 May in which Quilmes is almost as large as La Matanza in absolute value, with approximately 1/3 of its population. Whereas Quilmes *is* in the top position when scaled to relative per 100 000 inhabitants.

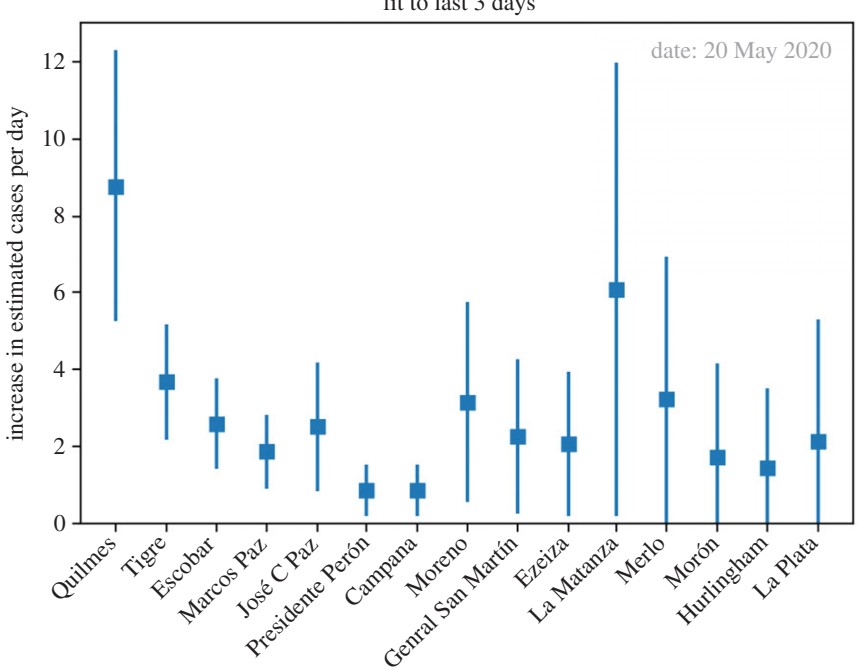

**Figure 5.** Slope of a linear fit to the cases per day estimation of last 3 days. The error bar corresponds to including the uncertainty in each per day estimation and in the slope determination in the fit. Also in this plot it is crucial to sort the districts according to the significance in this variable. In the figure, we see on 20 May Quilmes in the top position, indicating a potential early alarm for an outbreak, as was consequently confirmed by other indicators a few days later.

early outbreak detection implies a main challenge in these complex cases where detection and propagation block must be done when the first few cases are reported. In particular, the above-described early alarm for the outbreak occurred in Villa Azul allowing a fast response of the Health Care system team to mitigate and control its propagation to Villa Itatí.

Once the strict lock-down and isolation had been implemented, water and food supply were delivered by the social care team. People were not allowed to leave the house during the entire isolation. Active surveillance health teams started with a door-to-door symptoms monitoring. Those cases with clinical manifestation related to COVID-19 were tested. Confirmed cases were isolated

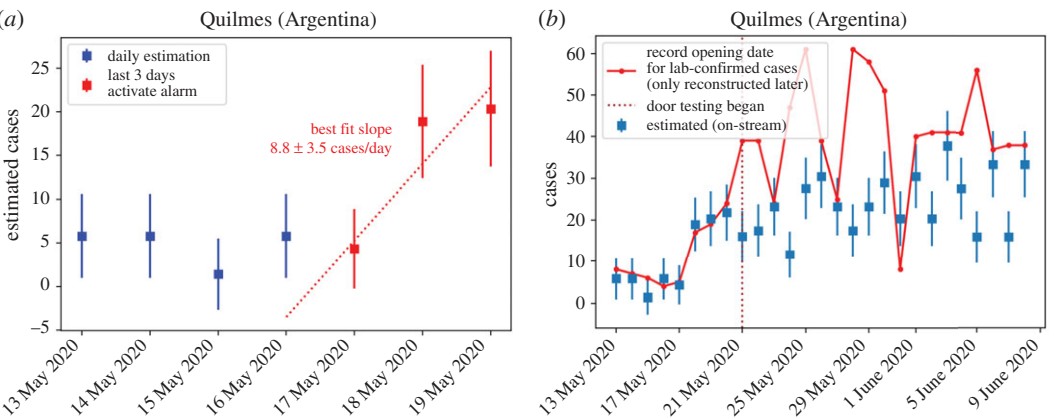

**Figure 6.** (*a*) Exact Early Outbreak Alarm visualization on 20 May for Quilmes, as available to the Health Care team in Buenos Aires Province. (*b*) The posterior picture including the laboratory-confirmed cases (solid red), the day that the Health Care system landed on Villa Azul to begin testing door to door, and a wider range of dates to capture the big picture of the case. The strict lock-down in Villa Azul with no entering or leaving permission lasted from 24 May to 8 June. As can be seen in the plot, during the door-to-door testing, the solid red line goes above and uncorrelated to the estimated cases by phone calls, as expected.

inside their houses in cases where this was possible (if there was an empty room for example) and in cases where it was not, the people were sent to an out-of-hospital centre.

## 5. Outlook and scope

The development of the mathematical model to estimate the number of COVID-19 cases was done in urgency and adapting it to the available data. There was no time to request changes in data acquisition or processing. Of course, the algorithm and the system can be improved in many directions. We discuss some of these features in the following paragraphs.

One of the major weaknesses in the algorithm is the large granularity, which corresponds to districts. District populations in the area are on average 500 000 people. This issue is translated in that the Early Outbreak Alarm stops working once the density of cases is such that there are more than a few outbreaks in each district. This happened in late June in PBA. In a future implementation, we are carrying out a workaround for this issue by obtaining a reliable address from the COVID-trained operator who takes the call. A more stable solution would be to obtain this information from the telephone company; however, regulations many times block this possibility.

On the other hand, the algorithm has a very important benefit that is its unbiasedness. Given that the COVID-line works 24 h a day, 7 days of the week and with a fair equal methodology all the time, the algorithm estimation does not rely on tests availability or overloaded testing facilities, among others. Of course, the system does have slight biases that may come—for instance—from different backgrounds due to different features in the districts, or seasonally social behaviour as months go by. Some of these biases may be solved by re-fitting the model once in a while, others by fitting different models in different regions.

Importantly, the algorithm provides information about the background calls that vary in space and time. Further studies could be done in order to understand and extract properties of the background, as it can be its seasonality, variations according to regions, to public announcements or news, etc.

The crucial point in the mathematical model is that it recognizes anomalies due to collective behaviours. Therefore, we find that the mathematical model and the Early Outbreak Alarm algorithms can be useful for many other epidemiological diseases—as for instance dengue—and other events such as natural catastrophes, among others. We are currently working on the improvement of this system in many aspects, also including machine learning algorithms, and these advancements will be published in a future work.

## 6. Conclusion

We have created a syndromic surveillance algorithm based on the correlation between phone calls to a COVID-line, districts population and reported cases. This algorithm works by understanding that

phone calls to a COVID-line are in part from non-infected people having similar symptoms (background) and in part from infected people (signal). By observing that background has to be proportional to district population, whereas signal proportional to reported cases, we have fitted our assumption. The coefficient of determination for Buenos Aires Province (PBA) is always $R^2 > 0.85$ for different samples, which indicates the robustness of our hypothesis. In addition, we have validated our model with real data.

In this paper, we have described the model, its estimations and how we compute their error bars. Also, it has been shown how the estimations, which are obtained in-stream, can be used to address Public Health policies without requiring to wait for laboratory results, which require many more days to converge. The algorithm worked in PBA from April to June, since during this time the COVID-trained call centre was not overloaded. Therefore, the estimation was relatively unbiased.

We have shown how this estimation can be used to create an Early Outbreak Alarm. Furthermore, we describe how the construction of indicators that have to do with daily cases, and daily increase of cases, can indicate outbreaks in advance. The relevant statistical variable in this case is the significance, since it is a real measure on how far from zero are the indicators. Importantly, this system can detect an outbreak and, in particular, we exemplify its application in the outbreak detection in Villa Azul (Quilmes).

The limitations to the Early Outbreak Alarm were discussed in this paper. Many of them arise because of the characteristics of the data available at the moment of its (urgent) development. We have pointed out many ways to improve its sensitivity and accuracy, on which we are currently working. This alarm would also be useful, not only for other epidemiological diseases but also for events that yield changes in collective behaviour, such as dengue epidemic, natural catastrophes or others.

The presented algorithm and mathematical model provide a helpful tool to prevent the spread of SARS-CoV-2 and could be useful in improving the measures discussed recently [23]. This algorithm has been one of the main tools in the PBA Health Care system dashboard during the epidemic, and its current and upgrade versions are still being used to track the epidemic and detect outbreaks.

Data accessibility. Number of phone calls each day from each district are available upon request.

Authors' contributions. E.A. has proposed the model and fitted the parameters; F.M. has contributed with the maths, the pre-processing of the data and the building of the team; S.C., E.G. and N.K. have performed the organization for collecting the data and the data selection and collection it self; D.O. has performed the epidemiologic analysis of the work.

Competing interests. D.O., S.C., E.G., N.K. and F.M. work for the the Buenos Aires Health Ministry and have technical and political responsibility within the Health Care Administration.

Funding. Research work of E.A. has been through project PICT-2018-03682 from ANPCyT. Research of F.M., S.C., E.G., D.O. and N.K has not been funded.

Acknowledgements. We thank the fantastic work performed by the 148 COVID call centre, in particular to R. Vaena, P. Rispoli and L. H. Molinari for useful conversations. E.A. and F.M. thank Dr I. Caridi for useful conversations. E.A. thanks CONICET, UNSAM and EasytechGreen for finantial and logistic support during this research, and CAF and F. Lamagna for supporting features and developments in the upgraded version.

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
