## [Peer Review File · Royal Society Open Science]

Review History

RSOS-202312.R0 (Original submission)

Review form: Reviewer 1

Is the manuscript scientifically sound in its present form?

Yes

Are the interpretations and conclusions justified by the results?

Yes

Is the language acceptable?

Yes

Do you have any ethical concerns with this paper?

No

Have you any concerns about statistical analyses in this paper?

No

Recommendation?

Accept with minor revision (please list in comments)

Comments to the Author(s)

This manuscript proposes an original and easily implementable protocol to anticipate epidemics outbreaks, in this case, linked to COVID-19. The method proves to lead to reliable and consistent results.

The authors base their proposal on the analysis of phone calls made by possible infected people to inform and learn about their symptoms. A close correlation between the dynamics of these calls and the development of the epidemics is the basis for the presented protocol.

I have a couple of questions for the authors, the first one regarding the model, the second one to the results.

1) The model uses the same factor θ for all chunks. This factor can be associated with an R eff of the disease but includes not only epidemiological aspects but sociocultural aspects the could make it necessary to consider different values for different chunks. How does the choice of a single mean (adjusted) parameter affect the sensitivity of the model?

2) The error bars seem to indicate that the model is less accurate in the most critical moments, but I could not find any discussion about this aspect. Do the authors know if the origin of this behaviour is due to the quality of data or to the intrinsic behaviour of the disease that at the peaks can more apparently display a broader latent period or clustered infectious linked to a single phone call?

Review form: Reviewer 2 (Nick Tyler)**Is the manuscript scientifically sound in its present form?**

Yes

Are the interpretations and conclusions justified by the results?

Yes

Is the language acceptable?

Yes

Do you have any ethical concerns with this paper?

No

Have you any concerns about statistical analyses in this paper?

No

Recommendation?

Accept with minor revision (please list in comments)

Comments to the Author(s)

This paper is a good and innovative contribution to the growing knowledge about the Covid-19 pandemic spread, and gives a very interesting approach to determining potential spikes in occurrence before they appear in the statistics.

My only comment for revision is that occasionally the English grammar could be corrected:

line 12ff Covid -19 is now considered to be aerosol based and thus I think 'infection' might be a better term here than "contagion"

line 22 "...System with a tool .."

line 30 I think "systems" rather than "perception"

line 38 "... main role in spatial..."

line 41 "...believes themselves to be..."

line 42 "...responds to people's questions..."

line 45 "...infection, they are instructed..."

line 46 "...such a syndromic..."; "...was used as an input..."

Page 4

line 7 "...caused by COVID-19..."

line 8 "... a COVID-specific phone line ..."

line14 This is not clear, but I think what is meant is that the number of staff operating the phone line increased so that all calls were answered (at least until late June), thus the data collected in that period were unbiased by whether or not the calls that were answered were proportionately reporting a Covid infection or some other Covid related enquiry. Unanswered calls of course would not have been able to have that assurance. If my understanding is correct, perhaps something like "The staffing of the COVID-19 phone line was increased as the epidemic spread..."

line 15 "...per day in March ..."

line 16 "... system was able to meet the demand, and ..."

line 22 " relatively stable."

line 25 " As users ..."

line 32 "...did not contain reliable information ..."

line 36 "...of the system"

Page 5

line 29 " ... as thresholds ..."

line 43 " allows the estimation of..."; "... but not the determination of ..."

line 53 "...especially the systematic ones, if present."

Page 6

line 8 "... registered on most occasions, in practice, ..."

line 11 " ... and Jun 26th only, because after this date te call centre was overloaded and all the calls could not be taken, ..."

line 13 "... For this period, ..."

Page 7

line 32 " ...for any two districts..."

line 42 "...that a potential outbreak ..."

line 49 " ... through this model is especially useful ..."

Page 8

line 3 " ... as occurred in ..."

line 40 "In this section, ..."

line 49 " Using the ..."

line 52 " ... on the rise."; "...epidemiological feature ..."

Page 9

line 6 "... and, on the other hand, it is also important..."

line 10 " As the implemented system is intended to be an Early Outbreak Alarm,..."

line 20 "As can be seen..."

line 25 " the number of estimated cases ... was below..."

line 33 This is not clear. "We have considered" suggests that you thought about it but did not actually do it. It is not clear from the text whether or not this was included in your method, so it would be helpful to be clear on that point.

line 44 "... has provided the PBA Health Care administration with ..."

line 51 "... would be as predicted ..."

line 52 "... controlled between mid April and mid June..."

Page 11

line 50 "... decided on a strict ..."

Page 12

line 25 "... no entering or leaving ..."

line 45 "Once the strict lockdown and isolation had been implemented, water ..."

Page 13

line 13 " District populations in the area are on average 500K..."

line 18 "reliable address..."; ..."operator who takes the call ..."

line 23 " 24 hours a day, seven days a week,..."

line 39 "...working on the improvement .."

line 48 " ... based on ..."

line 50 " ... in part on..."

line 51 "...and in part on ..."

Page 14

line 8 "In this paper, we have described the model ..."

line 18 "... the relevant statistical variable ..."

line 23 "... the limitations to ..."

line 23 "... discussed in this paper. Many of them arise because ..."

line 49 "...responsibility within ..." (or, if appropriate, "responsibility for"

Decision letter (RSOS-202312.R0)

Dear Dr Alvarez

On behalf of the Editors, we are pleased to inform you that your Manuscript RSOS-202312 "Estimating COVID-19 cases and outbreaks on-stream through phone-calls" has been accepted for publication in Royal Society Open Science subject to minor revision in accordance with the referees' reports. Please find the referees' comments along with any feedback from the Editors below my signature.

Please submit your revised manuscript and required files (see below) no later than 7 days from today's (ie 19-Feb-2021) date. Note: the ScholarOne system will 'lock' if submission of the revision is attempted 7 or more days after the deadline. If you do not think you will be able to meet this deadline please contact the editorial office immediately.

on behalf of Professor Matjaz Perc (Associate Editor) and Mark Chaplain (Subject Editor)
openscience@royalsociety.org

Associate Editor Comments to Author (Professor Matjaz Perc):

I concur with the positive recommendations. A couple of additional points:

- Please use phone calls, not with the hyphen.
- Related research is: Forecasting COVID-19, Front. Phys. 8, 127 (2020).
- The authors should please make their source code available for others to re-use
- In terms of actions for containment, a good set of rules was proposed for Europe in: An action plan for pan-European defence against new SARS-CoV-2 variants, Lancet 397, 469-470 (2021)

Reviewer comments to Author:

Reviewer: 1
Comments to the Author(s)

This manuscript proposes an original and easily implementable protocol to anticipate epidemics outbreaks, in this case, linked to COVID-19. The method proves to lead to reliable and consistent results.

The authors base their proposal on the analysis of phone calls made by possible infected people to inform and learn about their symptoms. A close correlation between the dynamics of these calls and the development of the epidemics is the basis for the presented protocol.

I have a couple of questions for the authors, the first one regarding the model, the second one to the results.

1) The model uses the same factor θ for all chunks. This factor can be associated with an R eff of the disease but includes not only epidemiological aspects but sociocultural aspects the could make it necessary to consider different values for different chunks. How does the choice of a single mean (adjusted) parameter affect the sensitivity of the model?

2) The error bars seem to indicate that the model is less accurate in the most critical moments, but I could not find any discussion about this aspect. Do the authors know if the origin of this behaviour is due to the quality of data or to the intrinsic behaviour of the disease that at the peaks

can more apparently display a broader latent period or clustered infectious linked to a single phone call?

Reviewer: 2

Comments to the Author(s)

This paper is a good and innovative contribution to the growing knowledge about the Covid-19 pandemic spread, and gives a very interesting approach to determining potential spikes in occurrence before they appear in the statistics.

My only comment for revision is that occasionally the English grammar could be corrected:

Page 2

line 12ff Covid -19 is now considered to be aerosol based and thus I think 'infection' might be a better term here than "contagion"

line 22 "...System with a tool .."

line 30 I think "systems" rather than "perception"

line 38 "... main role in spatial..."

line 41 "...believes themselves to be..."

line 42 "...responds to people's questions..."

line 45 "...infection, they are instructed..."

line 46 "...such a syndromic..."; "...was used as an input..."

Page 4

line 7 "...caused by COVID-19..."

line 8 "... a COVID-specific phone line ..."

line14 This is not clear, but I think what is meant is that the number of staff operating the phone line increased so that all calls were answered (at least until late June), thus the data collected in that period were unbiased by whether or not the calls that were answered were proportionately reporting a Covid infection or some other Covid related enquiry. Unanswered calls of course would not have been able to have that assurance. If my understanding is correct, perhaps something like "The staffing of the COVID-19 phone line was increased as the epidemic spread..."

line 15 "...per day in March ..."

line 16 "... system was able to meet the demand, and ..."

line 22 " relatively stable."

line 25 " As users ..."

line 32 "...did not contain reliable information ..."

line 36 "...of the system"

Page 5

line 29 " ... as thresholds ..."

line 43 " allows the estimation of..."; "... but not the determination of ..."

line 53 "...especially the systematic ones, if present."

Page 6

line 8 "... registered on most occasions, in practice, ..."

line 11 " ... and Jun 26th only, because after this date te call centre was overloaded and all the calls could not be taken, ..."

line 13 "... For this period, ..."

Page 7

line 32 " ...for any two districts..."

line 42 "...that a potential outbreak ..."

line 49 " ... through this model is especially useful ..."

Page 8

line 3 " ... as occurred in ..."

line 40 "In this section, ..."

line 49 " Using the ..."

line 52 " ... on the rise."; "...epidemiological feature ..."

Page 9

line 6 "... and, on the other hand, it is also important..."

line 10 " As the implemented system is intended to be an Early Outbreak Alarm,..."

line 20 "As can be seen..."

line 25 " the number of estimated cases ... was below..."

line 33 This is not clear. "We have considered" suggests that you thought about it but did not actually do it. It is not clear from the text whether or not this was included in your method, so it would be helpful to be clear on that point.

line 44 "... has provided the PBA Health Care administration with ..."

line 51 "... would be as predicted ..."

line 52 "... controlled between mid April and mid June..."

Page 11

line 50 " ... decided on a strict ..."

Page 12

line 25 "... no entering or leaving ..."

line 45 "Once the strict lockdown and isolation had been implemented, water ..."

Page 13

line 13 " District populations in the area are on average 500K..."

line 18 "reliable address..."; "...operator who takes the call ..."

line 23 " 24 hours a day, seven days a week,..."

line 39 "...working on the improvement .."

line 48 " ... based on ..."

line 50 " ... in part on..."

line 51 "...and in part on ..."

Page 14

line 8 "In this paper, we have described the model ..."

line 18 "... the relevant statistical variable ..."

line 23 "... the limitations to ..."

line 23 "... discussed in this paper. Many of them arise because ..."

line 49 "...responsibility within ..." (or, if appropriate, "responsibility for")

===PREPARING YOUR MANUSCRIPT===

===PREPARING YOUR REVISION IN SCHOLARONE===

- If you are requesting a discretionary waiver for the article processing charge, the waiver form must be included at this step.
- If you are providing image files for potential cover images, please upload these at this step, and inform the editorial office you have done so. You must hold the copyright to any image provided.
- A copy of your point-by-point response to referees and Editors. This will expedite the preparation of your proof.

- Ensure that your data access statement meets the requirements at <https://royalsociety.org/journals/authors/author-guidelines/#data>. You should ensure that you cite the dataset in your reference list. If you have deposited data etc in the Dryad repository, please only include the 'For publication' link at this stage. You should remove the 'For review' link.
- If you are requesting an article processing charge waiver, you must select the relevant waiver option (if requesting a discretionary waiver, the form should have been uploaded at Step 3 'File upload' above).
- If you have uploaded ESM files, please ensure you follow the guidance at <https://royalsociety.org/journals/authors/author-guidelines/#supplementary-material> to include a suitable title and informative caption. An example of appropriate titling and captioning may be found at https://figshare.com/articles/Table_S2_from_Is_there_a_trade-off_between_peak_performance_and_performance_breadth_across_temperatures_for_aerobic_scope_in_teleost_fishes_/3843624.

Author's Response to Decision Letter for (RSOS-202312.R0)

See Appendices A - C.

Decision letter (RSOS-202312.R1)

Dear Dr Alvarez,

It is a pleasure to accept your manuscript entitled "Estimating COVID-19 cases and outbreaks on-stream through phone calls" in its current form for publication in Royal Society Open Science.

COVID-19 rapid publication process:

We are taking steps to expedite the publication of research relevant to the pandemic. If you wish, you can opt to have your paper published as soon as it is ready, rather than waiting for it to be published the scheduled Wednesday.

This means your paper will not be included in the weekly media round-up which the Society sends to journalists ahead of publication. However, it will still appear in the COVID-19 Publishing Collection which journalists will be directed to each week (<https://royalsocietypublishing.org/topic/special-collections/novel-coronavirus-outbreak>).

If you wish to have your paper considered for immediate publication, or to discuss further, please notify openscience_proofs@royalsociety.org and press@royalsociety.org when you respond to this email.

on behalf of Prof Mark Chaplain (Subject Editor)
openscience@royalsociety.org

Appendix A

Dear Professors Lianne Parkhouse and Matjaz Perc,

We thank you for carrying on the manuscript considerations within the Royal Society Open Science Journal.

We have addressed all the Reviewers comments in the manuscript and replied in different letters to their reviews. We have also included all comments made by the Editorial Office as well.

We are including a highlighted version in which all changes proposed by Reviewer #1 and the Editorial office are in blue color. Since Reviewer #2 comments were mainly typos and English grammas suggestions, these are not highlighted.

In the 'clean' version of the manuscript all changes are in normal text.

With kind regards,

Ezequiel Alvarez
Daniela Obando
Nicolas Kreplak
Sebastian Crespo
Franco Marsico
Enio Garcia

Appendix B

We thank Reviewer for carefully reading the manuscript and for his/her very suitable questions and suggestions. We have addressed both questions below, and we have modified the manuscript accordingly.

Question #1:

The model uses the same factor θ_I for all chunks. This factor can be associated with an R_{eff} of the disease but includes not only epidemiological aspects but sociocultural aspects the could make it necessary to consider different values for different chunks. How does the choice of a single mean (adjusted) parameter affect the sensitivity of the model?

This is an important observation. In fact, it is true that θ_I includes sociocultural aspects and that it can vary along the different regions within a city. In particular θ_I variation can provide insightful information: in our case, since the COVID-line is for public health service, it provides us -among others- with a correlation of public and private health distribution. We do see a slight biased factor in the case estimation in those regions with higher economic level which have a preference for private health services, however this factor is maintained throughout time. Therefore, although the algorithm has a slight bias in the absolute value in these specific regions, it is faithful in the variations in time, and therefore it is still useful in tracking variations.

There is an interplay related to the amount of available data in either fitting the same θ_I for all regions or using different θ_I for each region. In the first case, one reduces the relative statistical uncertainty at the price of having eventual biases due to sociocultural variations. In the second case, one has larger relative statistical uncertainties, but one is sensitive to sociocultural variations in the predictions. In our work we have fitted the same θ_I for all regions, accepting that some sociocultural/economic biases are introduced in some regions. However, with enough data it is recommendable to fit different θ_I for different regions. Moreover, this can provide important information for further analyses.

We have completed the paragraph before Eq. 2 by discussing the above aspects through the possibility of using different fits for the θ 's parameters in different regions.

Question #2:

The error bars seem to indicate that the model is less accurate in the most critical moments, but I could not find any discussion about this aspect. Do the authors know if the origin of this behaviour is due to the quality of data or to the intrinsic behaviour of the disease that at the peaks can more apparently display a broader latent period or clustered infectious linked to a single phone call?

The error bars are in fact larger when the number of cases increase. Although it is likely that systematic sources due to quality of data appear in these peak periods, we have not considered them in this work. The effect we understand the referee is pointing out is due to the Poissonian component in the uncertainty, since it goes as \sqrt{N} and therefore the absolute uncertainty increases with N . However, the relative size of the error bar compared to the predicted value, that is \sqrt{N}/N , decreases as $1/\sqrt{N}$. Therefore, although larger N shows larger error bars, the fluctuations around its central value are smaller in terms of percentage of the prediction.

We have modified the manuscript accordingly and at the end of the paragraph including Eq. 2 we have added a discussion on the size of the error bars and their relationship to the absolute number of estimated cases.

We thank the Reviewer for pointing out these items which have contributed to improve the quality of the work. We hope to have fulfilled his/her inquiries with above answers.

Ezequiel Alvarez
Daniela Obando
Nicolas Kreplak
Sebastian Crespo
Franco Marsico
Enio Garcia

Appendix C

We thank Reviewer for carefully reading the manuscript and for his/her very useful comments and observations. We have addressed all suggestions and we have modified the manuscript accordingly.

The only point that may require an explicit answer is the following:

Page 9

line 33 This is not clear. "We have considered" suggests that you thought about it but did not actually do it. It is not clear from the text whether or not this was included in your method, so it would be helpful to be clear on that point.

The Reviewer is correct, we have replaced 'considered' -> 'chosen' that is what actually happened.

We thank the Reviewer for pointing out these items which have contributed to considerably improve the quality of the work.

Ezequiel Alvarez

Daniela Obando

Nicolas Kreplak

Sebastian Crespo

Franco Marsico

Enio Garcia